# White Light Emission by Simultaneous One Pot Encapsulation of Dyes into One-Dimensional Channelled Aluminophosphate

**DOI:** 10.3390/nano10061173

**Published:** 2020-06-16

**Authors:** Rebeca Sola-Llano, Ainhoa Oliden-Sánchez, Almudena Alfayate, Luis Gómez-Hortigüela, Joaquín Pérez-Pariente, Teresa Arbeloa, Johan Hofkens, Eduard Fron, Virginia Martínez-Martínez

**Affiliations:** 1Departamento de Química Física, Universidad del País Vasco, UPV/EHU, Apartado 644, 48080 Bilbao, Spain; rebeca.sola@ehu.eus (R.S.-L.); ainhoa.oliden@ehu.eus (A.O.-S.); teresa.arbeloa@ehu.eus (T.A.); 2Instituto de Catálisis y Petroleoquímica-CSIC, C/Marie Curie 2, Cantoblanco, 28049 Madrid, Spain; a.alfayatelanza@gmail.com (A.A.); lhortiguela@icp.csic.es (L.G.-H.); jperez@icp.csic.es (J.P.-P.); 3Department of Chemistry, Katholieke Universiteit Leuven, Celestijnenlaan 200F, B-3001 Heverlee, Belgium; johan.hofkens@kuleuven.be

**Keywords:** dye-guest encapsulation, zeolite, microporous aluminophosphates, one-pot synthesis, hybrid fluorescent system, white light emitter, FRET

## Abstract

By simultaneous occlusion of rationally chosen dyes, emitting in the blue, green and red region of the electromagnetic spectrum, into the one-dimensional channels of a magnesium-aluminophosphate with AEL-zeolitic type structure, MgAPO-11, a solid-state system with efficient white light emission under UV excitation, was achieved. The dyes herein selected—acridine (AC), pyronin Y (PY), and hemicyanine LDS722—ensure overall a good match between their molecular sizes and the MgAPO-11 channel dimensions. The occlusion was carried out via the crystallization inclusion method, in a suitable proportion of the three dyes to render efficient white fluorescence systems by means of fine-tuned FRET (fluorescence resonance energy transfer) energy transfer processes. The FRET processes are thoroughly examined by the analysis of fluorescence decay traces using the femtosecond fluorescence up-conversion technique.

## 1. Introduction

Hybrid organic-inorganic materials are recognized as promising systems for many applications in different fields such as optoelectronics, energy, and biomedicine [1,2,3,4,5]. Generally, the final properties are not a mere sum of the individual contributions of each moiety; new and improved features also emerge as a result of synergetic effects [1,5,6]. In this way, new materials with interesting features for innovative technologies can be produced.

White light emitters are one of those potential materials that are extremely interesting for lightning systems, necessary in technological devices, and present in everyday life. Hitherto, white light emission has been largely pursued to improve existing lightning technology [7,8,9,10,11,12,13,14,15]. In this regard, several approaches have been postulated. In particular, fluorescence resonance energy transfer, FRET, is a bioinspired process to develop new systems as light-emitting devices [16,17,18]. The control of energy transfer processes with the combination of proper dyes allows to tune the color of the resultant emitted light. The most common strategy to achieve white light emission consists of combining fluorophores with emission spectra in the three fundamental regions (blue, green, and red), emitting at well-defined intensities into a single host system able to impose a proper spatial distribution regarding intermolecular distances and molecular orientations [8,19,20,21,22,23]. Note that although purely organic compounds can allow a low-cost fabrication of white emitter devices, their major drawback is the aging, usually at different rates for each component [23,24]. However, encapsulated organic dyes within nanocavities, preferably inorganic frameworks, are sheltered from chemical-, photo- and thermo-degradation due to the protection provided by the porous host [25,26]. Microporous zeolitic materials are ideal candidates to be used as inorganic stable hosts, in particular microporous aluminophosphates based on AlO_4_ and PO_4_ alternate tetrahedral units [26].

Generally, optical applications based on the combination of organic fluorescent dyes with inorganic hosts require optically dense materials. In such cases, molecular aggregation should be completely avoided in order to preserve the optical characteristics of the components. The key to success in the design of these hybrid materials is the matching between dye molecular size and nanochannel dimensions. In this context, the best synthetic approach to reach a tight host-guest fit is via the crystallization inclusion method, in which the dye is embedded while the inorganic framework is being formed [26,27]. Thus, the opening dimension of the pores limits the size of the guest, and since the occlusion process does not involve diffusion, the guest dyes can be accommodated all along the host crystals, avoiding guest entangling at the pore entrance. However, this is not a trivial approach and several aspects should be considered before the encapsulation of organic dyes within inorganic channelled structures. The first condition when choosing an organic dye for this approach is that its chemical structure should not be very different from that of the organic structure-directing agents (SDA) that typically drive through a template effect the crystallization of zeolitic materials, usually amines or quaternary ammonium compounds. In particular, dyes should be able to bear positive charges to maximize interaction with the negatively-charged inorganic framework, enhancing their incorporation. Moreover, the dye must be soluble in an aqueous synthetic gel. For this reason, cationic organic dyes with amine groups are a preferred choice. Besides, a negative net charge should be generated in the zeolitic framework, i.e., by the isomorphic substitution of Al^3+^ by Mg^2+^ in aluminophosphate networks, to facilitate the entry of those cationic dyes.

In previous works, it has been demonstrated that the one-dimensional Mg-containing aluminophosphate MgAPO-11, with AEL structure, is a perfect host for the incorporation of many commercial dyes due to its adequate pore structure having non-intersecting one-dimensional (1D) channels (6.5 × 4 Å) and their special topology [28,29]. By the encapsulation of dyes through the crystallization inclusion method into this framework, not only is molecular aggregation avoided, but preferential alignment along the pores is also induced [27,30,31]. Moreover, the inorganic 1D-host MgAPO-11 offers a very rigid environment limiting molecular motions responsible for the non-radiative deactivation pathways to the ground-state, yielding an astonishing enhancement of fluorescence with respect to the dye in solution, particularly for dyes with flexible molecular structures. As a result, highly fluorescent hybrid materials with an anisotropic response to the linearly polarized light have been obtained using this particular host [31].

In fact, the encapsulation of different commercial fluorescent dyes into the 1D-nanochanneled inorganic host MgAPO-11 (Scheme 1) via a one-pot synthesis has already rendered hybrid materials with diverse optical properties such as (i) an optically switchable system by the simultaneous encapsulation of two chromophores with a complementary response to linearly polarized light [29]; and (ii) a red-emitting hybrid material with Second Harmonic Generation (SHG) properties under IR excitation by the confinement of a nonlinear optics (NLO) dye perfectly aligned within the channels [31].

Particularly in this work and as above-mentioned, a critical aspect to attain pure white light after excitation with UV light is to control the efficiency of energy transfer among the different co-adsorbed fluorescence emitters. Taking into account the requirements for an effective Försters Resonance Energy Transfer or FRET [32], acridine (AC), pyronin Y (PY), and hemicyanine LDS722 dyes (Scheme 1) were chosen to achieve a white light hybrid emitter. AC, with bluish-cyan emission, was selected as the primary energy-donor moiety due to its relatively high fluorescent quantum yield and its small dimensions that would provide a high donor-rate into the host. The AC emission spectrum overlaps well with the absorption band of the next energy-acceptor molecule, PY. Indeed, this AC-PY pair was set in a previous work as a suitable donor-acceptor pair for an efficient FRET process [29]. Finally, a hemicyanine-like dye, 4-[4-[4-(dimethylamino)phenyl]-1,3-butadienyl]-1-ethyl-pyridinium (LDS722), characterized by emission in the red region of the electromagnetic spectrum, is chosen as final acceptor dye in the FRET cascade. Although the fluorescence of this dye is rather poor in solution due to the inherent flexibility of its molecular structure, a highly improved red emission efficiency is reached by the great constraint imposed by the AEL nanopores [31].

Here, to achieve a bright emitting system with pure white light, a systematic variation of the relative proportion of the three dyes in the synthesis of the hybrid material is performed. The resultant materials are fully characterized by steady-state and time-resolved spectroscopic techniques. Experimental evidence of FRET processes among the dyes within the pores of the material are also provided via ultrafast spectroscopy experiments where the excited-state dynamics is investigated.

## 2. Materials and Methods

### 2.1. Synthesis

Mg-containing aluminophosphates (MgAPO-11) with the three dyes were prepared by hydrothermal crystallization method using phosphoric acid (Aldrich, Madrid, Spain, 85 wt.%), magnesium acetate tetrahydrate (Aldrich, Madrid, Spain, 99 wt.%,), aluminum hydroxide (Aldrich, Madrid, Spain), ethylbutylamine (EBA, Aldrich, Madrid, Spain), and the aforementioned dyes: AC (Aldrich, Spain 97%), PY (Acros Organics, Madrid, Spain, 75%), and LDS722 (Exciton through Lasing S.A, Madrid, Spain, laser grade, purity > 99%). All the compounds were used as received. The synthesis gels have a general molar composition of 0.2 MgO:1 P_2_O_5_:0.9 Al_2_O_3_:0.75 EBA:*x* dye:300 H_2_O, where *x* indicates the total amount of dye added into the gel. In this work, different dye-proportions were added to the gel, keeping constant the total amount of dye at *x* = 0.024 in order to not disrupt the final crystalline phase of the material. For example, a dye-proportion in the gel of 3AC:2PY:1LDS722 means that the molar fraction of AC dye in the synthesis gel is 1.5 times higher than the molar fraction of PY, and three times higher than that of LDS722. Considering that the total amount of dyes in the gel was set at 0.024, the proportion added for AC, PY, and LDS722 is 0.012, 0.008, and 0.004, respectively. The pH of the synthesis gels was between 4 and 5. The gels were heated under autogenous pressure for 24 h at 160 °C. The solid products were recovered by filtration, exhaustively washed with ethanol and water, and dried at room temperature overnight.

### 2.2. Characterization

X-ray powder diffraction (XRD) was used to determine the crystalline phase obtained; XRD patterns were collected with a Panalytical X’Pro diffractometer (Panalytical, Lelyweg, The Netherlands) using Cu Kα radiation.

The dye content within the solid products was photometrically determined using a double beam Varian spectrophotometer (model Cary 7000, Agilent Technologies, Madrid, Spain) after dissolving the solid material in 5 M hydrochloric acid and comparing the resulting solutions with standard solutions prepared from known concentrations of the dyes at the same pH value of the sample solutions.

The absorption spectra of the dye-containing MgAPO powder samples were recorded with a Varian spectrophotometer (model Cary 7000) detecting the reflected light through an integrating sphere. The respective spectra of the MgAPO powders synthesized under identical conditions but without dyes were recorded and subtracted from the sample signal to eliminate the scattering contribution to the absorption spectra. Emission spectra of the bulk powder were recorded in an Edinburgh Instruments spectrofluorimeter (FLSP920 model, Livingston, U.K.) in front-face configuration (40° and 50° to the excitation and emission beams, respectively) and leaned 30° to the plane formed by the direction of incidence and detection. The fluorescence lifetime decay curves of the bulk powder were measured with the time-correlated single-photon counting technique in the same spectrofluorimeter using a microchannel plate photomultiplier tubes (MCP-PMTs, Hamamatsu R38094-50) with picosecond time-resolution (~150 ps). The fluorescence lifetime (τ) was obtained after deconvolution of the instrumental response signal from the recorded decay curves by means of an iterative method. The goodness of the exponential fit was controlled by statistical parameters (chi-square, χ^2^, and analysis of the residuals).

Absolute photoluminescence quantum yields of the dye-containing MgAPO powders were measured in an integrated sphere coupled to this spectrofluorimeter. The absorbance at excitation wavelength was obtained by comparing the scatter signal of the dye-loaded hybrid material and a Teflon disk was used as a reference (with a diffuse reflectance of 100%).

An amplified femtosecond double OPA (optical parametric amplifier) laser system was used to provide excitation pulses of 395 nm. The power of the excitation beam was set to 300 µW (300 nJ/pulse, power density = 1697 mW/cm^2^, energy density/pulse = 1.70 × 10^−3^ Joule/(pulse × cm^2^), number of photons/(pulse × area) = 3.20 × 10^15^ photons/(pulse × cm^2^)), and the fluorescence light emitted from the samples was efficiently collected from the same side as the excitation beam using a concave mirror. The fluorescence was then filtered using a 420 nm long-pass filter to suppress the scattered light, directed and overlapped with a gate pulse (800 nm, ca. 10 µJ) derived from the regenerative amplifier onto a lithium triborate (LBO) crystal. By tuning the incident angle of these two beams relative to the crystal plane, the sum frequency of the fluorescence light and the gate pulse was generated. The time-resolved traces were then recorded by detecting this sum-frequency light while changing the relative delay of the gate pulse versus the sample excitation time. Fluorescence gating was done under magic-angle conditions in a time window of 50 ps. Monochromatic detection in heterodyne mode was performed using a PMT (R928, Hamamatsu, Shizuoka, Japan) placed at the second exit of the spectrograph mounted behind a slit. Optical heterodyne detection is a highly sensitive technique that is also used to measure very weak changes in absorption induced by a frequency-modulated pump beam. An additional bandpass filter 260–380 nm was placed in front of the monochromator in order to reject light from the excitation and the second harmonic of the gate pulse. The electrical signal from the photomultiplier tube was gated by a boxcar averager (SR 250, Stanford Research Systems, Sunnyvale, CA, USA) and detected by a lock-in amplifier (SR830, Stanford Research Systems). The prompt response (or instrumental response function, IRF) of this setup (including laser sources) was determined by detection of scattered light of the excitation pulse under identical conditions and found to be approximately 100 fs (FWHM: full width at half maximum). This value was used in the analysis of all measurements for curve fitting using iterative re-convolution of the datasets while assuming a Gaussian shape for the prompt response.

Global analysis of the fluorescence decays obtained at different wavelengths as a sum of exponentials allowed to obtain amplitude-to-wavelength spectra. When interpreting the features of those amplitude-to-wavelength spectra, one should keep in mind that no correction for the wavelength dependency of the sensitivity of the detection setup sensitivity (mixing crystal, filters, PMT) was applied. The samples were prepared in powder form and contained in a quartz cuvette. To improve the signal-to-noise ratio, every measurement was averaged 15 times at 256 delay positions where a delay position is referred to as the time interval between the arrival of the pump and gate pulses at the sample position.

## 3. Results and Discussion

The three dyes chosen to be simultaneously occluded into the MgAPO-11 inorganic host have already been encapsulated and individually analysed in the same framework. Acridine (AC)-loaded MgAPO-11 material (AC/AEL) has demonstrated a bluish-cyan emission under UV excitation light (absorption band, λ_abs_: 350–450 nm; emission maximum, λ_emiss_: 481 nm, and CIE “Commission Internationale de l’Eclairage” coordinates: 0.18,0.33) with a relatively high fluorescence quantum yield of 54% and a long fluorescence lifetime of 27 ns ascribed to the protonated species of AC monomers, ACH^+^ [29]. As green-emitting moiety, pyronin Y (PY) dye encapsulated into MgAPO-11 (PY/AEL) has revealed the characteristic green emission of the PY monomers (CIE under blue excitation light: 0.35,0.64), with an improved fluorescence quantum yield and a longer lifetime (ϕ_fl_: 0.29, τ_fl_: 4.2 ns) than that of PY in diluted aqueous solution (ϕ_fl_: 0.21, τ_fl_: 2 ns) [27]. Finally, LDS722 dye into AEL has shown red emission properties (LDS722/AEL, λ_fl_ = 670 nm, CIE under green excitation light: 0.69, 0.31) with a much higher quantum yield (ϕ_fl_: 0.55) than that of the corresponding one in aqueous solution (ϕ_fl_ ≤ 0.01) due to the rigidity imposed by the host matrix [31].

As stated before, white light emission in this AEL host containing a mixture of dyes must be assisted by a successive FRET energy transfer process among the fluorophores. Note here that FRET process from AC (donor) to PY (acceptor) into the MgAPO-11 matrix has already been explored, in which a high probability to encounter acceptor molecules in neighbouring channels was demonstrated [29]. In the present case, PY will be the energy acceptor in the first step, and the energy donor with respect to LDS722. Note here that even though LDS722 dye can be directly excited upon UV light through its Locally Excited “LE” state (Figure 1), the most efficient absorption band to obtain red emission is the broad band (450–680 nm) of LDS722, located in the green-visible region and assigned to an Intra Charge Transfer state “ICTS” [31]. This broad absorption band shows a good overlap with the emission band of PY/AEL and enables an efficient FRET process. However, the ICTS absorption band also overlaps with the fluorescence of AC/AEL, opening the possibility that the excitation energy of AC/AEL is directly transferred to the LDS722/AEL system. Therefore, the resultant emission of the present materials is expected to involve competitive FRET channels among the three dyes.

A requirement to obtain a white light-emitting material is optimization of the proportion of dyes occluded into AEL, and consequently, the amount and ratio added to the synthesis gel. It is worth mentioning that predicting the incorporation degree of each dye into the inorganic matrix by the crystallization inclusion method is not an easy task and it is even more difficult when a mixture of dyes is present in the synthesis gel. Apart from typical parameters such as the solubility and competition of the dyes’ uptake, which can favour or impede the accommodation of the others, the multiple available energy transfer processes involving the three occluded dyes make finding the proper proportions of each chromophore in the synthesis gel a trial-and-error process, which has been nevertheless based on the previous knowledge gained on the behaviour of the individual dyes when occluded in the aluminophosphate matrix. In this sense, a systematic variation of the relative proportion of the three dyes in the synthesis gel was performed. The most representative samples are listed in Table 1 and Figure 2. The emission color in all samples is confirmed by chromaticity experiments (CIE coordinates) and Color Temperature (CCT).

As a first attempt, dyes were added to the synthesis gel in equal proportions (Sample 1, *x* = 0.008 of each dye). CIE chromaticity coordinates (Table 1, Figure 2) indicate a resultant yellow emission in this material, showing a deficit mainly of blue components. In fact, the Color Temperature, CCT, obtained from CIE coordinates [33], with a numerical number of 3350 K, is categorized as “warm white” that is characterized by an orange to yellow-white pleasant color suitable for lighting bedrooms, living room, and restaurants.

Then, in the second attempt, the amount of blue-emitting AC dye in Sample 2 (first donor in the FRET cascade of our three-dyes system) was significantly increased in the gel (*x* = 0.017) and the amount of PY and LDS722 was reduced to the same extent (from *x* = 0.008 to *x* = 0.003), resulting in an AC:PY:LDS722 ratio of approximately 6:1:1 (Table 1). In this case, a much higher incorporation of LDS722 with respect to PY and even with respect to Sample 1 was achieved (Table 1). This experimental fact can be ascribed to the strong competition between PY and LDS722 dyes to be embedded into the MgAPO framework, the incorporation of PY being hindered by the presence of LDS722 in the gel. As a result, the fluorescent emission of Sample 2 shows CIE coordinates near the white color under UV excitation, but still with a slightly bluish hue (Figure 2). Accordingly, a relatively high CCT is derived (Table 1), being in the range of the so-called “cool white”, which resembles daylight with a bluish-white appearance usually employed when superior brightness is required such as in industrial areas, garages, retails, or art studios. However, neither of these two samples show a high fluorescence quantum yield (ϕ_fl_ ≤ 0.10).

In the next trial, Sample 3 in Table 1 with an AC:PY:LDS722 proportion in the gel of 4:2:1, the relative amount of PY with respect to LDS722 in the initial synthesis gel was increased with a double aim: (i) to provide a slightly higher contribution of green emission compared to Sample 2, and (ii) to improve the energy transfer from this green-emitting dye to the red-emitting entity. Indeed, as stated before, PY is supposed to show a dual behaviour both as an acceptor of the energy coming from AC and as a donor giving its energy to LDS722. Besides, the AC amount was slightly reduced (from *x* = 0.017 to *x* = 0.014) in an attempt to displace the emission-color out of the blue region. However, although not far from the white and with relatively higher efficiency (ϕ_fl_ = 0.10), in this case, the CCT number of 6070 K indicates a bluish hue of the white light, within the “cool light” range (Table 1, Figure 2).

Taking into account the latest results, in the next try, Sample 4, the amount of AC dye was kept constant (*x* = 0.014), but a slightly lower amount of PY with respect to Sample 3 was used, leading to a proportion of 3:1:1 in the initial gel. Its emission spectrum shows the characteristic bands of each dye, centered in the blue (480 nm), green (540 nm), and red (670 nm) regions of the visible spectrum with very similar intensities (Figure 3). In this way, pure white light emission under UV excitation light was finally achieved (see CIE coordinates in Table 1, Figure 2). According to its corresponding CCT number, a neutral white color is obtained, which is characterized by bright white light useful for the illumination of kitchens, bathrooms, offices, classrooms, or hospitals. Furthermore, this sample shows the highest emission quantum yield (around 17%) among all the samples described in this work (Table 1).

In sum, by (one-pot) adding three dyes to the synthesis gel with the following composition: 0.2 MgO:0.9 Al_2_O_3_:1.0 P_2_O_5_:0.75 EBA:0.014 AC:0.005 PY:0.005 LDS722:300 H_2_O, corresponding to a dye molar ratio of 3 AC:1 PY:1 LDS722, a pure white light-emitting material with the highest efficiency was achieved. In this sample, the estimated final amount of dyes loaded was 8.26 mmol of AC, 0.06 mmol of PY, and 0.55 mmol of LDS722 per 100 g of sample powder (Table 1), which corresponds to a relative dye proportion of 138:1:9, respectively. However, it is worth stressing that the quantification of the uptake of all dyes was not a trivial task due to the overlap between the absorption regions of the dyes in solution. Furthermore, the incorporation of each dye into the MgAPO-11 structure does not follow a linear correlation with the initial proportion added to the synthesis gel (Table 1). This fact is a consequence of the strong competition for getting occluded among the dyes, i.e., LDS722 obstructs the PY incorporation, as previously mentioned. The size of PY (perpendicular to the major molecular axis) is larger than that of AC and LDS722, what causes a tighter fit of PY in the AEL nanochannels, and hence a more difficult incorporation.

With the aim of discovering new materials with different dye-ratios to render white light emission of a high variety of white color appearance and compare their emission capacity, a new one, Sample 5, with a higher amount of LDS722 with respect to Sample 4, was prepared (3 AC:1 PY:2 LDS722 in the synthesis gel, Table 1). Note that the incorporation of the dyes at any proportion does not modify the AEL structure (Figure A1). In this sample, neutral white light emission (CCT = 4600 k) was also achieved. However, this sample showed a lower fluorescence quantum yield with respect to Sample 4 (Table 1).

To gain more insights into the FRET process involved among the three dyes, excited-state dynamics of the most representative samples (Samples 4 and 5) were studied. In this context, FRET process is confirmed by the differences found in the fluorescent lifetimes recorded for each dye in the presence of the other two dyes, with respect to their lifetimes when individually encapsulated into AEL (Table 2). Although it is not easy to discern and assign each lifetime obtained due to the overlapping of the emission bands of the dyes, a general trend was found (Table 2). In the first place, the significant decrease of the long fluorescence lifetime ascribed to the protonated species of AC (donor) from 27 ns in the absence of any acceptor to 18.7 ns and 16.20 ns for Samples 4 and 5, respectively, indicates a deactivation of the AC (donor) emission as a consequence of the energy transfer process. On the other hand, short lifetimes (<1 ns) recorded in both samples at this wavelength (λ_em_: 480 nm) could be attributed to the FRET process as these short contributions do not show up in the absence of acceptor(s) (vide infra).

Secondly, in Sample 4, the lifetime assigned to PY remains nearly unchanged as when it is individually occluded, likely due to a low participation in the energy transfer, this time from PY (as donor) to LDS722 (as acceptor). The lifetime ascribed to PY in Sample 5 undergoes a slight decrease of its characteristic lifetime in the AEL framework, together with the appearance of a short lifetime (Table 2), indicating that PY acts in this sample by transferring its excitation energy to a large extent to LDS722 as final acceptor dye. Regarding LDS722 lifetime, a slight decrease with respect to the sample with LDS722 alone is also found in both samples, reaching 2.4 ns (Table 2), likely ascribed to a slight change in its molecular conformation.

To better characterize FRET processes in these hybrid samples, fluorescence decay traces were recorded by femtosecond fluorescence up-conversion (100 fs resolution) on a 50 ps time window. Figure 4 displays the fluorescence decays obtained for Sample 4 on a 50-ps time scale detected at 450, 550, and 650 nm corresponding to the emission bands of AC/AEL, PY/AEL, and LDS722/AEL, respectively. The fit analysis retrieved an ultrafast component with a positive amplitude at 550 nm (decay component) and a negative amplitude at 650 nm (rise component). This suggests the presence of a kinetic process that leads to the depopulation of the locally excited state of the PY chromophore and population of the excited state of LDS722. Such a process is characteristic of energy transfer (FRET) and its fast rate indicates a highly efficient excitation energy transfer for the fraction of donor-acceptor involved occurring from PY to LDS722. Since the amplitude of this ultrafast component is roughly one-third of the longer component attributed to the fluorescence decay, we can estimate that for this sample, the fraction of the coupled donor-acceptor systems PY-LDS722 reaches 30%. Interestingly, the overall signal of the fluorescence decay of the AC moiety is rather low, suggesting an energy transfer occurring at a rate faster than the resolution of the setup (100 fs). Nevertheless, the weak fluorescence decays with an ultrafast and a fast component indicates a second energy transfer channel towards the PY or LDS722 moiety.

Figure 5 shows the fluorescence decays for Sample 5 (ratio 3:1:2), plotted on a 50-ps time scale in which the emission bands for AC/AEL, PY/AEL, and LDS722/AEL were recorded at 450, 550, and 650 nm. Similar to that of Sample 4, the fluorescence assigned to PY (550 nm) shows a decay component of 2.5 ps, whereas this appears as a rise component in the emission of LDS722 (650 nm). Such spectroscopic evidence points again to the presence of an energy transfer process from PY to LDS722 as acceptor and donor moieties. Moreover, considering the 60% contribution of this component to the overall decay (see Figure 5, 550 nm detection), we can state that a higher fraction of donor-acceptor systems is present in Sample 5 relative to Sample 4. This is, indeed, in line with the presence of double amount of LDS722 acceptor (ratio 3:1:2) when compared to Sample 4 (where the ratio was 3:1:1). Only a weak emission has been observed in the spectral emission region of AC/AEL (450 nm), indicating a strong quenching process due to FRET.

Figure A2 and Figure A3 show the time-resolved fluorescence emission spectra recorded for Samples 4 and 5, respectively. The spectra show the relative ratio between the emission intensities of the AC/AEL, PY/AEL, and LDS722/AEL constituents in 50 ps time window. It is clear that the fluorescence emission attributed to LDS722 (650–700 nm region) builds up in the first few picoseconds after excitation.

The fluorescence up-conversion experiments clearly indicate, at least for the PY-LDS722/AEL donor-acceptor pair, the presence of an efficient excitation energy transfer process. For the AC dye, two additional fluorescence decay time constants were observed, suggesting the presence of energy transfer channels towards PY and LDS722 within AEL.

The results presented above indicate that it is very difficult to predict the best synthesis conditions in terms of the dye-ratio in the initial gel. However, after optimization of the dye proportion, a solid white light hybrid emitter was achieved, which shows a relatively high fluorescence capacity of nearly 20%.

Finally, a photo-stability experiment was carried out for Sample 4 with the highest brightness (ϕ_fl_ = 0.17) and a CCT in the neutral white range (5170 K). The sample was irradiated with a continuous UV light (λ_exc_ = 355 nm and an irradiance of 2.75 mW/cm^2^) for different times. After 6 h of intense UV illumination, the sample lost less than 10% of its efficiency and changed its neutral white to cool white colour (CCT = 6400 K) because the red fluorescent LDS722 dye underwent a faster photobleaching process with respect to the blue acridine and green pyronine Y emissive dyes. The change in colour is depicted in Appendix A
Figure A4.

## 4. Conclusions

A new hybrid material has been designed herein by simultaneous encapsulation in appropriate amounts of three dyes, acridine (AC), pyronin Y (PY), and LDS722, with complementary spectroscopic bands emitting in the blue, green, and red regions of the visible spectrum, into the 1D-Mg-containing aluminophosphate framework with AEL structure, revealing an efficient white light emission under UV irradiation.

The final color is a consequence of successive partial energy transfer processes among the dyes, which can be modulated by the final uptake of each dye in the host matrix. Once the initial dye ratio is optimized in the synthesis gel, this straightforward synthesis through the one-pot occlusion of the chromophores offers an easy and fast approach to render a solid-state system with pure white light emission and an efficiency of around 20%. This type of hybrid material offers versatile white light emitter systems in which their brightness and emission color purity can be easily tuned. More importantly for future commercial applications, this material is easy and inexpensive to manufacture, and the process is not time-consuming since the incorporation of the dyes and synthesis of the hybrid material takes place in a unique step.

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
