# Peer review of "White Light Emission by Simultaneous One Pot Encapsulation of Dyes into One-Dimensional Channelled Aluminophosphate"

_nanomaterials, 2020, doi:10.3390/nano10061173_

Round 1
Reviewer 1 Report
In this work, the authors synthesized white light materials by combining of aluminophosphate with AEL-zeolitic type structure MgAPO-11 and three dyes with blue, green and red emission colors via in-situ synthesis method. Through rational selection of these three dyes with different molar ratios, white light for Sample-4 can be obtained with a relative high quantum efficiency of 0.17 and CCT of 5170 K. However, due to the complicated energy transfers among them and no linear correlation between the content incorporated into the channel and the initial proportion added, it is difficult for the authors to get the best conditions for synthesis of the white light materials. The manuscript can be published after the modifications. The authors should consider the following concerns.
1. In the “Materials and Methods” part, the authors mentioned the XRD, but through the manuscript, I didn’t see the XRD results. How did the effect of the doping contents (single or mixed dyes) on the structures of the materials?
2. From the UV/Vis spectra, how did the authors estimate the contents for each component in the mixed dyes system, for instance Sample-4?
3. The stability, such as photo-stability, should be carried out to see the overall and each component degradation, and eventually the effect on the CIE.
4 Some formats should be the same through the manuscript and some typos should be corrected, such as:
4.1 White Light or white light
4.2 color or colour
4.3 sample x or sample-x or Sample-x
4.4 Correlated Color Temperature or correlated color temperature
4.5 The titles in the refs use lowercase or capitalized words
4.6 Table 1, CCTe (K)
4.7 Ref 13, Eu3+, Tb3+
Author Response
We thanks reviewer #1 for his/her comments, it helps to improve the quality of the manuscript.
the point-by-point answers are below:
Review#1
In this work, the authors synthesized white light materials by combining of aluminophosphate with AEL-zeolitic type structure MgAPO-11 and three dyes with blue, green and red emission colors via in-situ synthesis method. Through rational selection of these three dyes with different molar ratios, white light for Sample-4 can be obtained with a relative high quantum efficiency of 0.17 and CCT of 5170 K. However, due to the complicated energy transfers among them and no linear correlation between the content incorporated into the channel and the initial proportion added, it is difficult for the authors to get the best conditions for synthesis of the white light materials. The manuscript can be published after the modifications. The authors should consider the following concerns.
In the “Materials and Methods” part, the authors mentioned the XRD, but through the manuscript, I didn’t see the XRD results. How did the effect of the doping contents (single or mixed dyes) on the structures of the materials?
The occlusion of dyes into the aluminophosphate framework by crystallization inclusion method does not modify the final AEL structure. A new XRD figure is now included in the supporting information where samples with and without dyes are compared.
2. From the UV/Vis spectra, how did the authors estimate the contents for each component in the mixed dyes system, for instance Sample-4?
As described in page 3 (lines 132-135), the real dye uptake of each sample was quantified spectrometrically, after dissolving a certain amount of the sample powder in hydrochloridric acid (5M). The absorption spectra of the samples are compared with the spectra of standard solutions prepared from known concentrations of the dyes in analogous conditions. The absorption spectra were recorded with a UV-Vis spectrophotometer (Varian, model Cary 4E).
However, the quantification for the AEL systems with three dyes simultaneously encapsulated is not a trivial task. In this case, the quantification of PY has been carried out as described above, but the determination of the amount occluded in the case of AC and LDS 722 dyes cannot be properly estimated with this standard procedure since at such acidic conditions the absorption bands of both dyes overlap in the UV-blue region. To shift the absorption band of the LDS 722 dye towards the green part of the Visible spectrum, and thus to avoid the overlapping with the ACH+ band, after dissolving the powder in 5M hydrochloric acid, EtOH solvent is added to prepare a 50:50 solution. Under these conditions (50:50 H2O:EtOH), the absorption band of LDS 722 is placed at 490 nm, red-shifted with respect to its absorption band in pure water, centered at 446 nm, which allows isolating at the same time the ACH+ band.
Dye content values are given throughout this work as mmol dye per 100 g sample powder
The stability, such as photo-stability, should be carried out to see the overall and each component degradation, and eventually the effect on the CIE.
The increase in the thermo and chemo stability of the organic compounds occluded into inorganic frameworks is well known and widely tested. In the manuscript, reference 25 is related with this topic.
In the revised version we have included a photo stability experiment (page 7 lines 343-349): Sample 4 is being irradiated under UV light (355 nm at 2.75 mW/cm2) for different times. The changes on the CCT are also detailed in Figure S4 included in the supporting information.
4 Some formats should be the same through the manuscript and some typos should be corrected, such as:
4.1 White Light or white light
4.2 color or colour
4.3 sample x or sample-x or Sample-x
4.4 Correlated Color Temperature or correlated color temperature
4.5 The titles in the refs use lowercase or capitalized word
4.6 Table 1, CCTe (K)
4.7 Ref 13, Eu3+, Tb3+
The formats and typo errors have been corrected throughout the manuscript
Reviewer 2 Report
The authors have developed “one pot” synthesis approach enabling to prepare white light emitting hybrid material by combining three dyes into inorganic Mg-aluminophosphate host. They also studied relevant properties of obtained materials by various spectroscopic techniques incl. time-resolved fluorescence providing information on FRET processes. The reported research is of high quality and it is timely due to search for tunable white light emitters, although 20 % efficiency is not the highest I can assume that authors are looking for conversion of near-ultraviolet LED radiation with their hybrid material. It would be worth of mentioning in the paper for which kind of light sources new materials are developed for. This rises the following question typical for any LED excited light source. How stable are the new hybrid materials (especially organic dyes) because of high UV-radiation flux and elevated temperature provided by LEDs? This is worth of separate study of course, but in the context of the present papers author can comment on this aspect as well based on their existing knowledge.
The term “UV illumination” is used by authors in several figure captions, but without specifying what is the source.
Lines 149-152: A description of absolute photoluminescence quantum yields is given. Later in the text, I did not any reference to these measurements?
A few technical comments.
Line 145: using a multichannel plate photomultiplier detector …
A multichannel plate is also detector, but it is used to detect VUV radiation or particles.
Line 165: x area) = 3.20 x 1015 photons/(pulse x cm2 )) and .
Is it correctly written?
I recommend to publish the manuscript after minor revisions.
Author Response
We thanks reviewer #2 for his/her comments, it helps to improve the quality of the manuscript.
The point-by-point answers are below:
Review#2
The authors have developed “one pot” synthesis approach enabling to prepare white light emitting hybrid material by combining three dyes into inorganic Mg-aluminophosphate host. They also studied relevant properties of obtained materials by various spectroscopic techniques incl. time-resolved fluorescence providing information on FRET processes. The reported research is of high quality and it is timely due to search for tunable white light emitters, although 20 % efficiency is not the highest I can assume that authors are looking for conversion of near-ultraviolet LED radiation with their hybrid material. It would be worth of mentioning in the paper for which kind of light sources new materials are developed for. This rises the following question typical for any LED excited light source. How stable are the new hybrid materials (especially organic dyes) because of high UV-radiation flux and elevated temperature provided by LEDs? This is worth of separate study of course, but in the context of the present papers author can comment on this aspect as well based on their existing knowledge.
As reviewer ·#1 has made also the same suggestion, and for this reason we have performed a photostability experiment consisting on the irradiation of sample 4 under UV at different times. The results are now included in the main text (page 7 lines 343-349) and a new figure is included in the supporting information (Figure S4).
The term “UV illumination” is used by authors in several figure captions, but without specifying what is the source.
The excitation wavelength is now detailed in the captions of Tables 1 and 2, and Figures 2,3,4 and 4. Since different excitation sources in the UV range have been employed (i.e. 350 nm for the fluorescence quantum yields, a band pass of 325-370 nm for the emission spectra, a laser at 400 nm and 395 nm for the fluorescence lifetimes and femto experiments, respectively, the general term UV irradiation is used along the main text.
Lines 149-152: A description of absolute photoluminescence quantum yields is given. Later in the text, I did not any reference to these measurements?
All the fluorescence quantum yields cited in the main text are referred to absolute photoluminescence quantum yields
A few technical comments.
Line 145: using a multichannel plate photomultiplier detector …A multichannel plate is also detector, but it is used to detect VUV radiation or particles.
We have changed its name into microchannel plate photomultiplier tubes (MCP-PMTs, Hamamatsu R38094-50)
Line 165: x area) = 3.20 x 1015 photons/(pulse x cm2 ) Is it correctly written?
It is now corrected in the text, it should be 3.20 x 1015 photons
Reviewer 3 Report
The authors report a series of hybrid materials built of Mg-aluminophosphate zeolite-type material with the incorporation of three different organic dyes, including blue emissive acridine, green emissive pyronine and red emissive LDS722. The authors successfully played with different ratios of three selected dyes within the zeolite-type framework achieving of various white-light emitting characteristics. This paper concerns a very important research aim related to the construction of efficient white light emitting materials, and it is a nice contribution to this research pathway.
Therefore, I recommend publication of this manuscript in the Nanomaterials journal. However, before publishing a few points should be addressed:
- The full name of LDS722 dye should be given.
- The authors stated that the positive charge of applied dyes is helpful for their incorporation within the inorganic framework. However, acridine is suggested to be used in the neutral form. Why?
- The important point, from the viewpoint of experimental work, is the optimization of the dye proportion within the inorganic matrix. The authors only shortly mentioned that the ratios used in the syntheses did not lead to the identical ratios in the final materials due to the preferential insertion of some dyes over others. This part should be discuss more precisely (the short statements such as LDS722 obstructs the PY incorporation appeared but they are not explained/discussed).
- Figures 4 and 5 contain the time-resolved spectra named fluorescence up-converson while the indicated excitation wavelengths seem to be of higher energy than the emission signals. This suggests rather a typical fluorescence, non an up-conversion mechanism. It will be good to explain this for the reader.
- In the conclusions, it will be good to point out the further challenges toward the enhancement of optical proprties (such as QY, brightness, emission color purity) in the similar hybrid materials.
Author Response
We thanks reviewer #1 for his/her comments, it helps to improve the quality of the manuscript.
The point-by-point answers are below:
Review#3
The authors report a series of hybrid materials built of Mg-aluminophosphate zeolite-type material with the incorporation of three different organic dyes, including blue emissive acridine, green emissive pyronine and red emissive LDS722. The authors successfully played with different ratios of three selected dyes within the zeolite-type framework achieving of various white-light emitting characteristics. This paper concerns a very important research aim related to the construction of efficient white light emitting materials, and it is a nice contribution to this research pathway.
Therefore, I recommend publication of this manuscript in the Nanomaterials journal. However, before publishing a few points should be addressed:
- The full name of LDS722 dye should be given.
- The authors stated that the positive charge of applied dyes is helpful for their incorporation within the inorganic framework. However, acridine is suggested to be used in the neutral form. Why?
The reason is the at acrinide dye (pKa= 5.1) is mainly protonated in the synthesis gel (pH of the gen is between 4 and 5) and consequently is incorporated in its positive form.
- The important point, from the viewpoint of experimental work, is the optimization of the dye proportion within the inorganic matrix. The authors only shortly mentioned that the ratios used in the syntheses did not lead to the identical ratios in the final materials due to the preferential insertion of some dyes over others. This part should be discuss more precisely (the short statements such as LDS722 obstructs the PY incorporation appeared but they are not explained/discussed).
As mentioned by the reviewer and in the manuscript (page 5 lines 235-238), the size of the three dyes determines their relative incorporation. AC and LDS722 have smaller dimensions (perpendicular to the major axis aligned with the channel direction) than PY; in fact, the latter is quite tightly confined within the elliptical 10MR channels of the AEL framework, and in consequence, its incorporation is more limited than those of the other two when they compete for getting confined during crystallization. As previously mentioned, this information was already included in the manuscript (lines 235-238 and lines 271-276).
On the other hand, a number of factors influence the incorporation of the dyes, including their solubility in the synthesis medium, their interaction with the framework walls and, in particular, the electrostatic interaction of their positive charge with the negative charged induced by the insertion of Mg in the framework; furthermore, some other difficult-to-predict factors also affect the incorporation of dyes, such as crystallization temperature, amount of Mg and spatial distribution, etc
- Figures 4 and 5 contain the time-resolved spectra named fluorescence up-converson while the indicated excitation wavelengths seem to be of higher energy than the emission signals. This suggests rather a typical fluorescence, non an up-conversion mechanism. It will be good to explain this for the reader.
The reviewer is right in the sense that excitation wavelength of the laser pulse is at higher energy than the emission. However, the term “up-conversion” refers to the method used to record spectra and fluorescence decay of the sample. In this method, the fluorescence is “gated” by a femtosecond pulse by producing the sum frequency between fluorescence and the “gate” laser pulse. The so obtained signal is “up-converted” and this is at higher energy that the fluorescence.
- In the conclusions, it will be good to point out the further challenges toward the enhancement of optical proprties (such as QY, brightness, emission color purity) in the similar hybrid materials.
The conclusion section has been extended.